# Feasibility Study of Strontium-Containing Calcium Phosphate Coatings on Micro-Arc Oxidized AZ31

**DOI:** 10.3390/ma18194509

**Published:** 2025-09-28

**Authors:** Satish S. Singh, John Ohodnicki, Abhijit Roy, Mitali Patil, Boeun Lee, Prashant N. Kumta

**Affiliations:** 1Department of Chemical & Petroleum Engineering, University of Pittsburgh, 815C Benedum Hall, Pittsburgh, PA 15261, USA; singhslu@gmail.com; 2Department of Bioengineering, University of Pittsburgh, Pittsburgh, PA 15261, USA; jmo30@pitt.edu (J.O.); or abjroy@gmail.com (A.R.); mitali.patil.1988@gmail.com (M.P.); boeun.eva.lee@gmail.com (B.L.); 3Center for Craniofacial Regeneration, University of Pittsburgh, Pittsburgh, PA 15261, USA; 4Department of Mechanical Engineering and Materials Science, University of Pittsburgh, Pittsburgh, PA 15261, USA; 5McGowan Institute for Regenerative Medicine, University of Pittsburgh, Pittsburgh, PA 15261, USA

**Keywords:** calcium phosphate, magnesium alloy, strontium, coating, corrosion, micro-arc oxidation

## Abstract

**Highlights:**

**What are the main findings?**

**What is the implication of the findings?**

**Abstract:**

Calcium phosphate coatings are known for their osteoconductive prowess. In this work, calcium phosphate coatings were studied on a model biodegradable magnesium alloy of AZ31, primarily to provide improved corrosion protection and, more importantly, to confer in vitro cytocompatibility to the AZ31 alloy. Correspondingly, an aqueous-based approach was developed to deposit Sr^2+^-substituted calcium phosphates on micro-arc oxidized AZ31. Micro-arc oxidation was used mainly as a pretreatment technique due to improved homogeneity and adhesion strength in comparison to the coatings formed by the traditionally used alkaline and acidic pretreatment. Calcium phosphate coatings with up to 11.5 mol. % Sr were formed on micro-arc oxidized AZ31 substrates. Despite observation of greater than the intended 10 mol. % Sr to the calcium phosphate coatings as measured within the measurement error, biphasic mixtures of dicalcium phosphate dihydrate and poorly crystalline hydroxyapatite were formed. Micro-arc oxidation treatment, nevertheless, provided a slight improvement in corrosion protection compared to uncoated AZ31. However, much-improved corrosion protection was provided by the calcium phosphate coatings prepared either with or without Sr^2+^. The calcium phosphate coatings prepared with Sr^2+^ were also observed to support improved MC3T3-E1 murine pre-osteoblast cell proliferation compared to the calcium phosphate coated substrates prepared without Sr^2+^.

## 1. Introduction

Calcium phosphate (CaP) coatings are very well studied and have been previously applied to biodegradable magnesium-based alloys to improve their corrosion resistance and biocompatibility [1,2,3,4]. A wide range of experimental techniques have also been used to deposit CaP coatings on biodegradable magnesium-based alloys. A few of the more commonly used techniques studied include plasma spraying, electrochemical-based techniques, sol-gel-based approaches, and aqueous-based methods [5,6,7]. Amongst the aqueous-based methods reported, biomimetic approaches and chemical conversion coatings are some of the more advantageous methods studied due to the capability to deposit various phases of CaPs in the presence of bioactive components under physiological conditions as well as the capability to deposit coatings of CaPs on 3D porous scaffolds [8,9,10]. Unfortunately, the CaP coatings deposited using aqueous-based approaches often exhibit relatively poor adhesion to the substrate and are also relatively inhomogeneous in composition and coverage [11].

Prior to the deposition of CaP coatings when utilizing aqueous-based approaches, substrate pretreatment is often required, especially when reactive biodegradable magnesium and magnesium-based alloys are considered [12,13]. Substrate pretreatment using both acidic and alkaline precursors has traditionally been explored, thus leading to the formation of CaP coatings, which exhibit the limitations previously described [14]. Furthermore, the use of electrochemical techniques as pretreatment methods prior to depositing CaP coatings using aqueous approaches has also been reported [15,16]. For coating applications that require a corrosion-resistant, smooth, and defect-free metal surface, pretreatments utilizing an electrochemical technique known as electropolishing have proven successful [17]. The oxide coatings formed after electrochemical treatment, specifically when using plasma electrolytic oxidation, also referred to as micro-arc oxidation (MAO), have been demonstrated to be homogeneous and provide improved adhesion strength in comparison to solution treatment in the absence of an electrical potential [18,19,20,21]. In addition, these oxidation-based MAO coatings can be customized and optimized to suit the needs of various alloys based upon the required corrosion rate by varying the chosen electrolyte and processing parameters during the oxidation process. This level of customization and refinement of post-processing allows for fine tuning and adjustment of specific alloy compositions that require specific elements for structural- and mechanical-based requirements [21,22,23]. Furthermore, cathodic deposition (CD) is a commonly used surface treatment to prepare surface protective layers but the use of the MAO process to generate coatings prior to CD treatment has been carried out to generate coatings with excellent friction and corrosion resistance [24]. The MAO technique has also been used to create a pore-sealing coating. Using a hydrothermal treatment of MAO-coated AZ31B alloy in a Na_3_PO_4_ solution, researchers were able to generate a pore-sealing coating in a novel one-step in situ growth process [25]. The deposition of coatings of CaPs during MAO treatment directly onto various biodegradable magnesium alloy substrates has also been demonstrated [26,27].

Although modified pretreatment approaches have been adopted to enhance the homogeneity and physical properties of CaP coatings, the deposition of cationic substituted CaPs in combination with these modified pretreatment approaches is yet to be studied in detail. Previous work studying Sr^2+^ substitution in CaP and micro-arc oxidized coatings on biodegradable magnesium alloys demonstrated that coatings containing Sr^2+^ supported enhanced osteoblast progenitor cell proliferation and differentiation towards a mature osteoblast phenotype [11,28]. A similar trend in increased cell proliferation and differentiation was also observed on strontium phosphate conversion coatings deposited on magnesium alloys [29,30]. Additionally, strontium-containing CaP coatings produced by other methods have yielded similar results and have showcased its positive effect on both cytocompatibility and corrosion resistance [30,31]. Furthermore, there are many reports on Sr-substituted CaP coatings in the literature [32]. This manuscript is not aimed at providing a comprehensive review on Sr-substituted CaP coatings. Furthermore, the goal of this manuscript is not directed at generating a particular phase of CaP coatings either. The current study is aimed primarily at documenting the use of MAO as a pretreatment technique and the efficacy of this pretreatment step to generate CaP coatings in undoped and doped forms.

Therefore, in the current study, the use of MAO as a pretreatment technique prior to depositing CaP coatings by an aqueous-based approach on a biodegradable model magnesium alloy prepared with 3% Al and 1% Zn (AZ31) was studied. This alloy was chosen as a model system specifically for its commercial availability, and it has been widely studied as a control alloy since it is known for possessing superior corrosion resistance, biocompatibility and cytocompatibility as compared to pure Mg [22]. The CaP coatings formed were prepared either with or without Sr^2+^ and no post heat treatment was utilized to form a specific CaP phase. A thorough characterization of the coatings generated was performed to determine their phase and elemental compositions. Furthermore, the capability of these coatings to provide corrosion protection to the magnesium alloy and to additionally support osteoblast progenitor cell proliferation was also assessed.

## 2. Materials and Methods

### 2.1. Substrate Preparation

A 0.81 mm thick foil of a Mg alloy containing 3% Al and 1% Zn (AZ31) was acquired from Alfa Aesar (Ward Hill, MA, USA). Square-shaped substrates, approximately 1 cm in length, were cut from the foil and were the subjected to acid etching in 3% HNO_3_ diluted in ethanol followed by rinsing in acetone, as previously described [8,11]. The etched substrates were then polished using SiC polishing paper with up to 4000 grit (Allied High Tech Products Inc., Compton, CA, USA). After polishing, the substrates were sonicated in acetone and stored in fresh acetone until further pretreatment using micro-arc oxidation (MAO).

### 2.2. Micro-Arc Oxidation Pretreatment

Prior to performing MAO treatment on the cleaned etched, and polished AZ31 substrates, electrical contacts with the cleaned AZ31 substrates were prepared. A conductive silver paste (AI Technology, Princeton, NJ, USA) was applied to the back of the AZ31 substrates. A piece of electrical wire was then embedded in the silver paste and the paste was allowed to set overnight at room temperature in a dry atmosphere. After 24 h, an epoxy resin (EpoxiCure, Buehler, Lake Bluff, IL, USA) was applied to the back of the substrate to provide insulation to the electrical contact. The epoxy was allowed to cure prior to performing any further experiments. The MAO electrolyte solution was prepared using 8 g L^−1^ Na_2_SiO_3_ (Alfa Aesar, Ward Hill, MA, USA), 6 g L^−1^ Na_3_PO_4_, and 4 g L^−1^ KF (Sigma Aldrich, St. Louis, MO, USA) [33]. It should be noted that the composition is original, although the components are typically used in MAO pretreatments as referenced on magnesium alloys such as AZ31. The concentrations were used and selected primarily based on the affinity of Mg to phosphate and silicate and fluoride ions. The general concept being that, with the application of high voltage, the arc generated would oxidize the surface of the substrate with possible generation of Mg-oxide, Mg-hydroxide, Mg-phosphate, and Mg-silicate on the surface of the AZ31 alloy. Correspondingly, the ratio of (silicate)/(phosphate)/(fluoride) was taken as 2:1.5:1 relying on the highest electronegativity of F^−^, followed by PO_4_^3−^ and then SiO_4_^4−^. The solution was stored at 4 °C prior to use for the deposition process. For the duration of the deposition of MAO coatings, the electrolyte solution was kept chilled. Following the preparation and storage of the electrolyte solution, the insulated substrates were immersed into the electrolyte solution while connected to a power supply (Gen2400W, TDK-Lambda, Neptune, NJ, USA). Once fixed, a voltage of approximately 350 V was applied. The coatings were deposited for 5, 10, or 20 min periods. At the end of the coating process, the substrates were then rinsed in deionized water and ethanol and were subsequently allowed to dry at room temperature.

### 2.3. Calcium Phosphate Coating Formation

After depositing the MAO coatings on AZ31 substrates, the 10 min MAO-pretreated substrates were immersed into one of two acidic solutions containing various concentrations of CaCl_2_·H_2_O, SrCl_2_·2H_2_O, and Na_2_HPO_4_ (Sigma Aldrich, St. Louis, MO, USA). Justification for depositing CaP coatings on only 10 min MAO-pretreated samples is provided in Section 3.1. The detailed concentrations used for generating the coatings are illustrated in Table 1. The pretreated substrates were kept in these solutions for 6 h on a platform shaker. The coating solution was kept at 60 °C for the entire duration of the CaP coating deposition process. The pH of the solution was not specifically monitored since the goal was not to generate a particular phase of CaP, which is very pH-dependent, but rather CaP coatings exhibiting single or multiple phases.

### 2.4. Characterization of Coatings

The coatings formed on the MAO-pretreated AZ31 substrate surfaces were carefully removed by scraping. The phase composition and crystallinity of the coatings deposited were then characterized using powder X-ray diffraction (X-Pert PRO Cu Kα λ = 1.5418 Å, with X’celerator detector, Malvern Panalytical, Westborough, MA, USA). During the measurements, the X-ray generator was operated at 45 kV and 40 mA, and the data was collected between 2θ values of 10 and 45°. Subsequent to X-ray diffraction analysis, Fourier-transform infrared spectroscopy (FT-IR, Nicolet 6700, Thermo Scientific, Waltham, MA, USA) was also performed on the collected powders and compared to commercially obtained hydroxyapatite (Sigma Aldrich, St. Louis, MO, USA) and dicalcium phosphate dihydrate (Sigma Aldrich, St. Louis, MO, USA) to analyze and validate the chemical linkages. The chemical species present in the X-ray identified the phase composition of the coatings deposited. The scans were performed using a spectral resolution of 2 cm^−1^. An average of 64 scans was reported for the data collected within a range of 400–4000 cm^−1^. The elemental composition of the coatings deposited was determined by dissolving 10 mg of the powders collected in 10 mL of a 3% HNO_3_ solution and analyzing the dilutions of these solutions using inductively coupled plasma optical emission spectroscopy (ICP-OES, iCAP duo 6500, Thermo Fisher Scientific, Hillsboro, OR, USA). Known standards were used for Ca, Mg, Sr, and P to generate the respective calibration curves. The microstructure of the coatings formed with varying elemental composition was also analyzed. Coated substrates were sputter-coated with palladium using a Cressington 108a sputter coater (Cressington, Watford, UK). The sputtered samples were then analyzed using scanning electron microscopy (SEM, Philips XL30, Thermo Fisher Scientific, Hillsboro, OR, USA) utilizing a spot size of 3 at an operating voltage of 10 kV.

### 2.5. Electrochemical Corrosion Experiments

Electrochemical characterization studies were performed using the CHI 604A instrument (CH Instruments, Inc., Austin, TX, USA). A 3-electrode cell set-up was used, with the prepared AZ31 substrates, an Accumet Ag/AgCl reference electrode (Fischer Scientific, Waltham, MA, USA), and a platinum wire serving as the working, reference, and counter electrodes, respectively. Hank’s balanced salt solution (HBSS), with a pH of 7.4, was used as an electrolyte solution to simulate in vitro conditions. All electrochemical testing was performed at 37 °C. The working electrodes for the electrochemical measurements were prepared by using a silver epoxy paste to establish contact at room temperature between the sample and a copper wire. The region of contact was then insulated at room temperature using a non-conducting epoxy resin such that only one face of the alloy (the one upon which contact was not established) would be exposed to the media. Open-circuit potential was run prior to testing to establish the electrochemical stability of the samples in the media. Electrochemical impedance spectroscopy (EIS) studies were carried out at open-circuit potential employing a sinusoidal amplitude of 10 mV over a frequency range of 100 kHz to 0.01 Hz. Experiments to obtain the Tafel plots were carried out from a voltage range of approximately −1.6 V to −1.2 V at a scan rate of 1 mV/s. All electrochemical corrosion experiments and subsequent analysis were performed on representative samples following repetition to demonstrate the accuracy of the synthesis of the micro-arc oxidation and the aqueous-immersion-based deposition approach to generate the undoped and doped CaP coatings on the model biodegradable magnesium alloy.

### 2.6. Mouse Preosteoblast Cell (MC3T3-E1) Culture

The murine preosteoblast cell line MC3T3-E1 was obtained from ATCC (Manassas, VA, USA). The cells were cultured at 37 °C in 5% CO_2_ and 95% relative humidity in minimum essential medium alpha (α-MEM, Gibco, Grand Island, NY, USA) containing 10% fetal bovine serum (FBS) and 1% penicillin streptomycin (P/S, Gibco, Grand Island, NY, USA). The cells collected after the third passage were used in all experiments and were seeded at a density of 50,000 cells per substrate.

### 2.7. Live/Dead Staining and SEM Imaging

After seeding the MC3T3-E1 cells directly on either coated or uncoated AZ31 substrates and culturing for 3 and 7 days in growth media, live/dead staining was used to evaluate cell proliferation (Invitrogen, Waltham, MA, USA, Live/Dead staining kit). Live and dead cell counts of the representative live/dead stained samples were measured using the watershed and analyze particle functions in ImageJ software version 1.53t. The samples were rinsed twice with phosphate-buffered saline (PBS) and were then incubated in PBS containing the live/dead stain for 40 min while protected from light. Prior to imaging using an Olympus CKX41 fluorescent microscope (Olympus Corporation, Center Valley, PA, USA) the staining solution was removed and replaced with PBS. The samples were then fixed using 2.5% glutaraldehyde and were subjected to ethanol dehydration. After sputter coating with palladium, they were imaged using scanning electron microscopy, as previously described. Images were taken of live/dead stained MC3T3-E1 preosteoblast cells and subsequent cell count measurements were performed on representative samples following repetition.

### 2.8. Statistical Analysis

All the experiments and subsequent data analysis conducted in this manuscript were performed on representative samples following repetition to demonstrate the accuracy of the synthesis of the MAO and the aqueous-based deposition approach to generate the undoped and doped CaP coatings on the model biodegradable magnesium alloy of AZ31. GraphPad Prism version 9 and Origin version 2023 were used to generate all the plots presented in this study.

## 3. Results and Discussion

### 3.1. Pretreatment Characterization

Pretreatment is often required when using solution-based approaches to deposit CaP-based coatings specifically on biodegradable magnesium alloy substrates [1,12]. In contrast to alkaline- or acidic-based pretreatment protocols, MAO coatings are specifically of interest due to their rapid deposition, homogeneity, and excellent adhesion to metallic substrates [26,27]. In the current work, MAO treatments were performed on polished AZ31 substrates for a period of ten minutes in an electrolyte solution containing 8 g L^−1^ Na_2_SiO_3,_ 6 g L^−1^ Na_3_PO_4_, and 4 g L^−1^ KF. As indicated in the experimental section, a potential of 350 V was applied for the duration of the coating process. At the end of the ten-minute period, the substrates were removed from the electrolyte solution and rinsed in ethanol and deionized water prior to further use. The microstructure of the coatings formed was analyzed using scanning electron microscopy and is depicted in Figure 1b and compared to bare polished AZ31 in Figure 1a. The characteristic highly porous structure of an MAO coating (Figure 1b) is clearly observed and devoid of any scratches seen on the bare polished AZ31 substrate (Figure 1a) [21,23,34]. Previous studies have been performed to reduce the porosity observed in micro-arc oxidized coatings, thereby enhancing corrosion protection using various approaches, such as optimizing the current density, working voltage, and frequency, as well as the addition of particulates and organic calcium salts during the coatings process [20,21,23,35,36]. The pores detected in the current work were generally much less than 5 µm in diameter, which was the largest size, and as small as ~1 µm. As expected, a relatively homogeneous coating was obtained. The deposition of the coatings for extended periods of 20 min while maintaining the voltage applied at 350 V was also studied. However, little or no difference in the microstructure and electrochemical corrosion protection was observed. In contrast, 5 min MAO pretreatment coatings exhibited significantly decreased corrosion protection as compared to 10 min MAO-coated AZ31Therefore, the etched and polished AZ31 substrates subjected to MAO for a period of ten minutes were, thus, used in all further experiments as described here.

### 3.2. Coating Characterization

The MAO-coated substrates, as indicated in the experimental section, were then immersed in solutions containing Ca^2+^, Sr^2+^, and PO_4_^3−^ kept at 60 °C for a period of 6 h while placed on a platform shaker (Table 1). At the end of this time period, the substrates were rinsed in ethanol and deionized water and were allowed to dry at room temperature prior to further use. The phase composition and chemical linkages of the species present in the CaP coatings formed on MAO-coated AZ31 from 4000 cm^−1^ to 450 cm^−1^ (Figure 2) and 1350 cm^−1^ to 450 cm^−1^ (Figure 3) were analyzed using FT-IR. The data collected for commercially available pure hydroxyapatite and pure dicalcium phosphate dihydrate, obtained from Sigma Aldrich, is also illustrated in Figure 2a,b, respectively, as well as Figure 3a,b, respectively. In comparing the spectra for hydroxyapatite and dicalcium phosphate dihydrate, broad OH- vibrational bands in the ranges of ~3532 cm^−1^ to ~3472 cm^−1^ and ~3255 cm^−1^ to 3162 cm^−1^, a distinct H-O-H vibrational band at ~1645 cm^−1^, and a broad H-O-H vibrational band at 1722 cm^−1^ were detected in the spectra collected for dicalcium phosphate dihydrate, which were not detected in the spectra for hydroxyapatite (Figure 2a,b). These bands, found only in the collected dicalcium phosphate dihydrate spectra, are indicative of the adsorbed water molecules that are characteristic of dicalcium phosphate dihydrate. Additional PO_4_^3−^ vibrational bands that are characteristic of only dicalcium phosphate dihydrate are found at ~518 cm^−1^, ~574 cm^−1^, 659 cm^−1^, ~982 cm^−1^, 1050 cm^−1^, and ~1115 cm^−1^. Dicalcium phosphate dihydrate characteristic HPO_4_^2−^ vibrational bands are found at ~781 cm^−1^, ~870 cm^−1^, ~1131 cm^−1^, and ~1200 cm^−1^ (Figure 3b). The distinct hydroxyapatite PO_4_^3−^ vibrational bands are detected at ~467 cm^−1^, ~557 cm^−1^, ~600 cm^−1^, ~962 cm^−1^, ~1016 cm^−1^, and ~1086 cm^−1^, whereas the distinct hydroxyapatite OH- vibrational band is detected at ~630 cm^−1^ (Figure 3a,b).

The pattern collected for the coatings formed without Sr^2+^ (MAO-CaP) closely resembled that of dicalcium phosphate dihydrate, although the dicalcium phosphate dihydrate characteristic PO_4_^3−^ bands at ~1050 cm^−1^ and HPO_4_^2−^ band at ~1131 cm^−1^ are absent. Instead, the presence of additional PO_4_^3−^ bands at ~1020 cm^−1^, ~599 cm^−1^, ~559 cm^−1^, and the cluster of bands around ~467 cm^−1^ indicate the presence of a secondary hydroxyapatite phase (Figure 3c). Interestingly, despite the addition of 10% Sr to the coating solution, a similar FT-IR pattern to that obtained without Sr^2+^ was observed (Figure 2d and Figure 3d). However, it is important to note that the stretch of bands at ~467 cm^−1^ is more distinct and of greater intensity in the CaP coating with Sr^2+^ (MAO-SrCaP) as compared to the CaP coating without Sr^2+^ (MAO-CaP), thus indicating the increase in the secondary hydroxyapatite phase (Figure 3c,d). Furthermore, neither the characteristic hydroxyapatite PO_4_^3−^ band at ~962 cm^−1^ nor the OH- band at ~630 cm^−1^ are present in either of the coatings, with or without Sr^2+^. The absence of the PO_4_^3−^ band at ~962 cm^−1^ and the OH- band at ~630 cm^−1^ from both CaP coatings, with and without Sr^2+^, in addition to the decreased intensity of the PO_4_^3−^ band at ~559 cm^−1^, suggests a weakly crystalline structure of the secondary hydroxyapatite phase.

Based on the FT-IR data collected, it appears that dicalcium phosphate dihydrate is the predominant CaP phase formed in the CaP coatings deposited on the MAO-treated AZ31 substrates with a minor phase of weakly crystalline hydroxyapatite. This is not surprising since the formation of dicalcium phosphate dihydrate coatings on Mg alloys using similar coating processes have been previously reported [37,38]. In addition, dicalcium phosphate dihydrate and hydroxyapatite are the only CaP phases which are stable under aqueous conditions [39]. Furthermore, during the deposition process, even if there was some dissolution of the underlying AZ31 substrate, this would result in a pH higher than the physiological pH of 7.4. Under these conditions, it is only hydroxyapatite that would be stable and, no other phases of CaP. If the pH were to fluctuate below the physiological pH, it is possible that there could be trace amounts of monetite and octacalcium phosphate (OCP) not detected in the FTIR. The goal of this manuscript, as mentioned earlier, is to study the influence of MAO pretreatment on formation of CaP coatings. More in-depth studies would be needed to ascertain the specific phases of CaP formed. The deposition of β-tricalcium phosphate (β-TCP) on Mg alloys using aqueous approaches has also been, however, reported in the literature [11,40]. Although unstable under aqueous conditions, the ionic substitution of Ca^2+^ ions by Mg^2+^ ions in the β-TCP structure has also been shown to stabilize the β-TCP phase [41,42].

To confirm the previously observed phase composition and determine the crystalline nature of the deposited coatings by varying the solution composition, X-ray diffraction was also performed (Figure 4). The collected X-ray diffraction data was analyzed, and the peaks of the deposited coatings were matched to hydroxyapatite (JCPDS 09-0432) and dicalcium phosphate dihydrate (09-0077) reference spectra and denoted by a triangle and star, respectively (Figure 4a,b). Similar to the results observed using FT-IR, dicalcium phosphate dihydrate peaks were detected in the X-ray diffraction data collected for the CaP coatings prepared without Sr^2+^ (Figure 4a). However, additional dicalcium phosphate dihydrate characteristic peaks at ~24.5° 2θ and ~31.5° 2θ were only present in CaP coatings without Sr^2+^ (MAO-CaP), suggesting the increased amount of dicalcium phosphate present (Figure 4a). Three relatively broad peaks between 30 and 35° 2θ were observed in the X-ray diffraction data for both deposited coatings, indicative of the presence of poorly crystalline hydroxyapatite, thus validating the findings of FT-IR seen in Figure 2 and Figure 3. Addition of 10 mol. % Sr^2+^ into the coating solution resulted in additional hydroxyapatite characteristic peaks observed at 35.6° 2θ and 39.5 ° 2θ, as well as a decrease in the dicalcium phosphate dihydrate peak at ~21 ° 2θ, thus indicating an increase in hydroxyapatite and a decrease in dicalcium phosphate dihydrate within the CaP coating with Sr^2+^ (MAO-SrCaP) (Figure 4b).

To further investigate the influence of Sr^2+^ on the CaP coatings formed, a Direct Derivation Method (DDM) semi-quantitative phase analysis was performed to estimate the phase composition of the hydroxyapatite and dicalcium phosphate dihydrate (Figure 4c). The DDM semi-quantitative phase analysis determined that CaP coatings without Sr^2+^ (MAO-CaP) are primarily dicalcium phosphate dihydrate and consist of 39.8% hydroxyapatite and 60.2 % dicalcium phosphate dihydrate. Addition of Sr^2+^ into the deposited coating (MAO-SrCaP) results in a coating consisting primarily of hydroxyapatite (59.4%) with dicalcium phosphate dihydrate (40.6%) as the secondary phase. In summary, a biphasic mixture of dicalcium phosphate dihydrate and poorly crystalline hydroxyapatite was formed on MAO-coated AZ31 substrates. In addition, the variation in solution composition with up to 10% Sr resulted in a biphasic mixture consisting of hydroxyapatite as the primary phase present within the deposited coating.

Previous work has also shown that relatively large amounts of Sr^2+^ can be substituted for Ca^2+^ in the crystal structure of various CaPs due to their similar ionic radii [43]. Therefore, it is not surprising at all that the CaP coating phase composition was not influenced by the addition of up to 10% Sr to the coating solution. To confirm the presence of Sr^2+^ in the CaP coatings formed, an elemental analysis of the powders collected after the coating process was performed, shown in Table 2. As expected, Sr^2+^ was not detected in the coatings prepared without Sr^2+^ and a Ca/P ratio (1.24) closer to 1, that of dicalcium phosphate dihydrate, rather than 1.67, that of hydroxyapatite, was observed. The presence of Mg^2+^ detected likely comes from the AZ31 substrate due to the changing pH to likely alkaline values during deposition, and likely substitution of Mg for Ca, the (Mg + Ca)/P ratio of 1.43 closer to that of 1.67 for hydroxyapatite is observed. With respect to Sr^2+^-containing coatings, a slightly larger value than 10 mol. % Sr^2+^ was detected, thereby confirming the presence of Sr^2+^ despite the similar phase composition upon addition of Sr^2+^ concentration in the coating solution. A Ca/P ratio of 1.21 and a (Ca + Sr)/P ratio of 1.54 with a (Ca + Sr + Mg)/P ratio of 1.8 greater than 1 was once again observed.

It should be noted that the XRD analysis shows no direct evidence of Mg substitution within the deposited coating. Direct evidence would only present itself if enough Mg substitution occurred to either cause a peak shift or result in the formation of another phase. The XRD data does not show evidence of either of the two. Additionally, Mg substitution within the CaP phases has been widely reported. Thus, the deposition process that occurs in parallel with the simultaneous possible corrosion of the AZ31 substrate is the source of any Mg and serves to validate justification for Mg substitution within the deposited CaP phase. The primary focus of this paper is to illustrate the influence of MAO pretreatment on formation of CaP phases with and without Sr. Thus, the presence of Mg and Sr in ICP analysis would indicate likely substitution in the CaP phase although existence as amorphous secondary phase is possible. Absence of any such chemical moieties containing Mg and Sr in the FTIR does therefore indicate that ionic substitution is likely. Peak shifts to lower angles further indicate substitution of Sr. However, quantitative evaluation of the extent of Sr substitution in the CaP phase has not been conducted in the present work since the focus was not to elucidate the influence of MAO to quantitatively generate any specific or different phases. The goal was only to illustrate how MAO pretreatment is effective in generating CaP phases that serve to provide corrosion protection. It is seen in Table 2 that Mg is detected due to the likely dissolution of Mg from the porous MAO coating of AZ31 substrate into the CaP coating. Sr substitution is also validated by the slight peak shifts observed, as well as the absence of additional Sr-based phases in the XRD data for Sr-substituted CaP coating. The presence of Mg and Sr substituting for Ca could synergistically affect the peak shifts in the XRD pattern, warranting further study, which will be conducted in the future. The slightly larger amount of Sr seen in the coating could arise due to possible instrumental error of ±2% relative standard deviation (RSD) error indicating 11.5 mol. % of Sr. Additionally, dissolution of Ca into the solution due to the basicity of the depositing solution occurring with dissolution of Mg from the AZ31 substrate could also change the cation-to-anion ratio. This, combined with the dual phase nature of the coating indicated by XRD and FTIR showing presence of brushite and hydroxyapatite, validates the different ratios of Ca/P, ((Ca + Sr)/P) and ((Ca + Sr + Mg)/P) mentioned earlier. The loss of Ca and presence of higher amount of Sr observed in the coating is reflected in the different ratios. However, as mentioned above, the primary focus of the paper is to illustrate the influence of MAO coating to generate various CaP phases. A more in-depth study will be conducted to ascertain quantitative phase differences in the future.

After confirming the presence of Sr^2+^ in CaP coatings, the influence of coating composition on the morphology of the CaP coatings was also evaluated. In Figure 5a,b, the microstructure of the CaP coatings deposited on MAO-coated AZ31 without Sr^2+^ is illustrated. In general, two distinct morphologies were observed. Rather elongated and flat plate like particles, characteristic of dicalcium phosphate dihydrate, and substantially smaller spherical particles of hydroxyapatite were both detected [44]. In Figure 5c,d, the microstructure of the CaP coatings deposited on MAO-coated AZ31 with Sr^2+^ is illustrated. Despite the addition of Sr^2+^, similar particle morphology was observed in comparison to the coatings prepared without Sr^2+^. The larger platelet morphology is characteristic of the brushite monoclinic phase also documented by the authors (see reference [11]), while the spherical particle morphology is akin to that of poorly crystallized hydroxyapatite.

In summary, biphasic mixtures of dicalcium phosphate dihydrate and hydroxyapatite were deposited on MAO-coated AZ31 substrates. It was also observed that approximately 11.5 mol. % Sr^2+^ was incorporated into these CaP coatings. Addition of Sr^2+^ within the CaP coatings resulted in higher amounts of hydroxyapatite and a biphasic mixture consisting primarily of hydroxyapatite, as confirmed by FT-IR and XRD. Additionally, the achievable mol. % Sr^2+^ incorporated within the deposited CaP coating reported herein is significantly higher than what has been previously reported in similar studies on Sr^2+^-containing CaP coatings [11,28,29,30,31]. Nonetheless, the incorporation of Sr^2+^ is likely to influence both the solubility and cytocompatibility of CaPs. As a result, Sr^2+^-containing CaP coatings may provide improved corrosion protection and support improved cell proliferation.

### 3.3. Electrochemical Characterization

In order to assess the corrosion protection capability of Sr^2+^ incorporated CaP coatings, the electrochemical characteristics of the coatings were analyzed using electrochemical impedance spectroscopy (EIS) and potentiodynamic polarization (PDP) and Tafel extrapolation. All the electrochemical experiments were conducted in Hank’s balanced salt solution (HBSS) (pH 7.4) maintained at 37 °C. All the conducted Tafel extrapolations are collectively illustrated in Figure 6, and all conducted EIS Nyquist interpretations, including theoretically fitted interpretations using representative equivalent circuit models, are illustrated in Figure 7. Figure 8 depicts the equivalent circuit model used for each treatment system and their corresponding calculated charge transfer resistance.

Micro-arc oxidation of the AZ31 alloy (MAO) led to a slight improvement in corrosion potential compared to uncoated AZ31 (AZ31) but significantly lowered the corrosion current density by a whole order of magnitude (see Figure 6b). Therefore, despite the highly porous microstructure observed after MAO (Figure 1), the MAO treatment of the AZ31 substrate leads to a slightly improved corrosion potential and a significantly reduced current density in comparison to an untreated AZ31 substrate, suggesting possible passivation and protection towards aqueous corrosion.

Deposition of CaP coatings with (MAO-SrCaP) or without Sr^2+^ (MAO-CaP) incorporation on MAO-coated AZ31 substrates, however, significantly improved the corrosion potential and slightly lowered the electrochemical corrosion current densities compared to MAO-coated AZ31 substrate. A near 60 mV improvement in electrochemical corrosion potential was observed for the Sr^2+^-incorporated CaP coatings, while a near 70 mV improvement was observed for CaP coatings without Sr^2+^. In contrast, the Sr^2+^-incorporated CaP coatings saw a more significant drop in current density than the CaP coatings without Sr^2+^ in comparison to the MAO-coated AZ31 substrate. Similar studies utilizing immersion coatings of Sr^2+^-doped CaP coatings on pre-treated AZ31 reported electrochemical corrosion current values ranging from 0.23 to 0.54 µA/cm^2^. These electrochemical corrosion current values are higher than the reported 0.0718 µA/cm^2^ for Sr^2+^-containing CaP (MAO-SrCaP) coatings. It is important to note that the aforementioned study used a pretreatment method which consisted of AZ31 immersion in Na_2_HPO_4_, followed by heat treating at 350 °C or 400 °C, as compared to the MAO pretreatment [11]. Thus, the chosen pretreatment method has a major effect on the overall corrosion resistance of the deposited Sr^2+^ containing CaP coating. In summary, the Tafel extrapolations demonstrated that CaP coatings enhance the electrochemical corrosion protection in comparison to the MAO-coated AZ31 substrates. Interestingly, Sr^2+^ presence in CaP coatings does not provide a substantial improvement in corrosion potential but lowers the current density considerably compared to CaP coatings without Sr^2+^.

Nyquist interpretations from EIS experiments were used to further assess and confirm the influence of the presence of Sr^2+^ in CaP coatings deposited on MAO-coated AZ31 on conferring corrosion protection to the AZ31 substrate. These plots are depicted by the black dotted lines in Figure 7a–d and the corresponding charge transfer resistance values calculated were obtained by fitting the Nyquist curves to an equivalent circuit (see Figure 8). Much unlike the potentiodynamic polarization (PDP) measurements, the EIS experiments are non-destructive and enable the acquisition of data that depicts the potential of the treatment conditions to provide a protective barrier to the underlying AZ31 substrate. In parallel with the current density obtained from the Tafel extrapolation curves, the MAO treatment significantly improved the charge transfer resistance of the substrate nearly seven-fold (from ~540 Ω to ~3685 Ω), as indicated by an increase in the diameter of the semi-circle and the corresponding intercept resistance on the real Z′ axis. Therefore, as indicated by the lower current density and corresponding higher charge transfer resistance value, despite the porous microstructure of MAO coatings, the process of MAO treatment passivates the AZ31 substrate significantly, and thus offers greater protection against corrosion. Also, in parallel with the current density, the CaP coatings containing Sr^2+^ have a greater charge transfer resistance (~5160 Ω) than CaP coatings without Sr^2+^ incorporation (~3970 Ω), as shown in Table in Figure 8. Therefore, as indicated by the lower current density and higher charge transfer resistance, Sr^2+^ incorporation into the CaP coatings leads to increased passivation and, thus, provides improved corrosion protection, despite the lower corrosion potential values of ~−1.375 V for MAO-SrCaP containing Sr^2+^ compared to ~−1.36 V for MAO-CaP devoid of Sr^2+^.

Additional analysis of the various MAO treatment conditions was performed by theoretically fitting the collected experimental EIS data of each coating system to the equivalent circuit models shown in Figure 8a, with the modeled Nyquist plots represented by a solid red line in Figure 7a,b. For both the bare and MAO-coated AZ31, two constant-phase element (CPE) loops were observed to be required in the equivalent circuit. The two CPE loops highlight that neither the oxide layer present on the bare AZ31, the MAO coating, or the AZ31 substrate acted as a perfect capacitor, thus implying the presence of a double layer. However, the addition of the CaP and SrCaP coating deposited by immersion of the MAO-coated AZ31 in solutions containing calcium salts, strontium salts, and sodium phosphate to generate the respective coatings required the addition of a third CPE loop to properly model the obtained experimental data. The corresponding charge transfer resistance values of each material system shown in Figure 8e further corroborate and showcase the effectiveness of the MAO treatment and the generation of the subsequent CaP and SrCaP coatings to provide an increase in corrosion resistance via an increase in charge transfer resistance of each system in HBSS.

In agreement with the experimentally obtained Nyquist plots, the MAO treatment increases the charge transfer resistance within the equivalent circuit of the bare AZ31 substrate from 541 Ω to 3687 Ω. An additional slight increase in the charge transfer resistance of 285 Ω is observed when depositing a secondary CaP coating on the MAO-coated AZ31 giving a charge transfer resistance of 3972 Ω. However, the incorporation of Sr^2+^ into the secondary CaP coating results in a significant increase in the charge-transfer resistance by 1474 Ω for a total charge transfer resistance value of 5161 Ω. Thus, these analyses suggest that the incorporation of Sr^2+^ within the CaP coating leads to an overall increase in corrosion protection as compared to non-Sr^2+^-containing CaP coatings, albeit the specific mechanism contributing to how Sr^2+^ provides the improvement is unknown. It is possible that Sr^2+^ incorporation and substitution into the apatite structure, which leads to the increased formation of highly stable hydroxyapatite and decreased formation of metastable dicalcium phosphate dihydrate (Figure 4), could render the CaP phase more stable in HBSS, thereby providing additional protection. Further investigation, however, into the effects of Sr^2+^ and the contribution of the Sr^2+^ ions on the solubility of the CaP coating and infiltration amount and depth within the MAO coating is warranted, which will be assessed in future studies planned.

As previously mentioned, numerous combinations of pretreatment and Sr^2+^-doped CaP coating deposition methods on Mg-based alloys have been utilized in efforts to increase the overall corrosion rate and cytocompatibility of the scaffold. Previous studies conducted by the authors utilizing a similar immersion solution method on AZ31 pre-treated via Na_2_HPO_4_ immersion and heat treatment up to 400 °C (see reference [11]) resulted in the deposition of a Sr^2+^-doped CaP coating with incorporation of up to 4.6 mol. % Sr and electrochemical corrosion current density of 0.46 µA/cm^2^. Unlike the corrosion potentials findings reported herein, a more negative shift in corrosion potential in conjunction with increasing corrosion current density was observed when coatings were formed within immersion solutions containing increased amounts of Sr^2+^ and heating during pretreatment to 400 °C. Additionally, the corrosion current density was doubled for CaP coatings deposited in a solution containing 10 mol. % Sr^2+^ and heat-treated to 400 °C as compared to 350 °C; however, this was not observed throughout all coating solutions tested. Singh et al. also found that decreased pretreatment temperatures and increased mol. % Sr^2+^ within the coating solution resulted in larger amounts of hydroxyapatite present within the deposited coating and consequentially increased corrosion protection [11]. Thus, the AZ31 pretreatment method plays a complex role on the deposited CaP coating and overall attainable corrosion protection. Furthermore, the heat treatment up to 400 °C required for the Na_2_HPO_4_ pretreatment can inadvertently alter the materials and mechanical properties of the underlying AZ31 alloy and may not be suitable for all Mg-based alloys. The current work which highlights using the same immersion solution coating method on a MAO pre-treated AZ31 substrate leads to increased Sr^2+^ incorporation within the biphasic hydroxyapatite and dicalcium phosphate dihydrate (~11.5 mol. %) and decreased electrochemical corrosion current density (0.072 µA/cm^2^). Thus, the current findings on Sr^2+^-containing CaP coatings deposited on MAO pre-treated AZ31 in comparison to Na_2_HPO_4_ and heat treatment-based pretreatment clearly highlights the effect of the pretreatment method on the deposited CaP coating.

Additional immersion solution-based methods used to deposit Sr^2+^-containing CaP coatings on pure Mg without a pretreatment have been reported [29,30]. Chen et al. (see reference [29]) immersed Mg substrates into a pH 3 adjusted solution containing 0.1 M Sr(NO_3_)_2_ and 0.06 M NH_4_H_2_PO_4_ at temperatures ranging from 40 °C to 80 °C. Coatings deposited at a solution temperature of 80 °C consisted of mainly Sr-Apatite, Mg(OH)_2_, MgHPO_4_, and Mg_3_(PO_4_)_2_, and exhibited the lowest corrosion current density (~7 µA/cm^2^) out of all the tested conditions. Park et al. (see reference [30]) immersed pure Mg substrate in a solution containing NaH_2_PO_4_·2H_2_O and varying amounts of Ca(NO_3_)_2_·4H_2_O and Sr(NO_3_)_2_, ranging from 0 to 2 mol. % Sr^2+^, at 80 °C for 3 h. The Sr^2+^-doped CaP coating deposited from immersion in a solution containing 2 mol. % Sr^2+^ at 80 °C consisted of dicalcium phosphate dihydrate and strontium phosphate and exhibited an improved electrochemical corrosion current density of 5.208 µA/cm^2^. Additionally, just like the herein-reported charge transfer resistances, Park et al. [30] also observed an overall increase in charge transfer resistance with increased mol. % Sr^2+^ within the coating solution. Even though the coatings reported by Park et al. and Chen et al. [29,30] increased the overall corrosion resistance of the Mg alloy, their reported corrosion current density values are higher than the values reported for the Sr^2+^-containing CaP coating (MAO-SrCaP) reported herein (0.072 µA/cm^2^) and thus provides less corrosion protection. Additional studies assessing the effectiveness of Sr^2+^-containing coatings on ZK60 Mg-based alloy deposited by MAO (see reference [28]) and immersion coating (see reference [31]) have been reported. Interestingly, Lin et al. elected to incorporate the Sr^2+^ directly into the MAO coating by adding Sr(OH)_2_ to the working electrolyte solution and did not deposit any secondary coatings like the work reported herein [28]. The Sr^2+^-containing MAO coating on ZK60 exhibited a drastic decrease in corrosion current density (0.151 µA/cm^2^) as compared to bare ZK60 (11.82 µA/cm^2^), albeit still not as low as the corrosion current density reported herein. An increase in resistance to polarization was also observed for Sr^2+^-containing coatings as compared to coatings without Sr^2+^. However, Lin et al. were only able to incorporate up to 0.228 mol. % Sr^2+^ within the deposited MAO coating, whereas the Sr^2+^-containing CaP coatings (MAO-SrCaP) reported herein consisted of ~11.5 mol. % Sr^2+^ [28]. Lastly, Makkar et al. (see reference [31]) hydrothermally deposited a biphasic coating consisting of dicalcium phosphate dihydrate and Sr^2+^-doped (1.06 at. % to 4.55 at. %) CaP on a non-pretreated ZK60 Mg-based alloy. The authors did not report any EIS or PDP results; however, the biphasic Sr^2+^-doped CaP coating significantly decreased corrosion as compared to bare ZK60 via hydrogen gas evolution analysis. No assessment of the effect of Sr^2+^ on corrosion resistance of the CaP coating was reported by Makkar et al. [31].

The above comparison of reported electrochemical and immersion methods used to deposit Sr^2+^-doped CaP coatings on pretreated and non-pretreated Mg-based alloys clearly highlights the positive effect Sr^2+^ plays on increasing corrosion resistance compared to undoped CaP coatings, as well as the effect pretreatment of the substrate has on the phase composition of the deposited coating and resulting corrosion resistance. Ultimately, the corrosion resistance provided by the deposited coating is directly correlated with the stability of the coating within the given environment and, thus, coatings consisting of a more stable phase of CaP will generally provide enhanced corrosion resistance. Hence, the increased corrosion resistance of the Sr^2+^-containing CaP coating (MAO-SrCaP) presented herein is likely in part due to the increased amount of highly stable hydroxyapatite and decreased amount of highly soluble dicalcium phosphate dihydrate as compared to CaP coatings without Sr (MAO-CaP). In conclusion, the reported AZ31 coating method consisting of a MAO pretreatment followed by a Sr^2+^-containing CaP coating deposited by immersion solution coating yielded the highest corrosion protection compared to the other hydrothermal, electrochemical, and pretreatments methods discussed herein. The current method also enabled incorporation of the highest mol. % Sr^2+^ within the deposited coatings compared to the other mentioned methods.

### 3.4. Cell Proliferation

MC3T3-E1 mouse preosteoblast cells were seeded directly on the MAO- and CaP-coated AZ31 substrates. The cells were cultured on these substrates for up to one week in growth media. Live/dead staining was performed after 3 and 7 days of culture, where live cells were stained green and dead cells were stained red (Figure 9 and Figure 10). The live and dead cell count of each representative live/dead image after 3 and 7 days of culture was quantified using ImageJ software (Table 3 and Table 4). After 3 days of culture, 402 live cells and 3 dead cells were observed as expected on tissue culture plastic, used as a positive control, while either only 3 live cells and 8 dead cells were detected on the MAO-coated AZ31 (Figure 9a,b and Table 3). On the other hand, 104 live cells and 18 dead cells were observed on the CaP coating without Sr^2+^ (MAO-CaP) and 182 live cells and 12 dead cells were observed on the CaP coating with Sr^2+^ (MAO-SrCaP). The incorporation of Sr^2+^ within the CaP coating (MAO-SrCaP) resulted in an 8.6% increase in live to dead cell count as compared to non Sr^2+^ containing CaP coatings (MAO-CaP). However, the number of live cells imaged for both these conditions was much lower than that observed on the tissue culture plastic at the same culture time.

After 7 days of culture, 670 live cells and 6 dead cells were measured on the tissue-culture plastic and an increase in live cell density was observed on the tissue-culture plastic as again expected in comparison to the earlier timepoint (Figure 10a and Table 4). With respect to MAO-coated AZ31, 248 dead cells stained in red were detected (Figure 10b and Table 4). For both CaP-coated AZ31 conditions, on the other hand, an increase in live cell density was observed (Figure 10c,d). Similarly, many more live rather than dead cells were detected for both CaP-coated conditions (Table 4). Interestingly, it should be noted that the coatings prepared with Sr^2+^ were observed to support nearly double the cell proliferation in comparison and an overall 7.2% increase in live to dead cells compared to those prepared without Sr^2+^ at the later time point. This is also consistent with the increased charge transfer resistance registered by the Sr^2+^ containing CaP coatings on MAO-treated AZ31 discussed above, indicative of the improved corrosion protection, which also contributes to better cell attachment.

However, the number of live cells observed on the coated samples was much lower than that observed on tissue culture plastic. Lastly, there was a very minimal decrease in % live cells from 3 days of culture to 7 days of culture for the CaP coatings with Sr^2+^ (MAO-SrCAP) and without Sr^2+^ (MAO-CaP), suggesting that the cytocompatibility of the coatings does not change with time. Future quantitative cytotoxicity assays will be performed to verify these qualitative and semi-quantitative findings.

After performing live/dead imaging, the cells were fixed, and the samples were prepared for imaging using scanning electron microscopy. The images collected on the CaP-coated samples after 7 days of culture prepared with and without Sr^2+^ are illustrated in Figure 11. As with the results obtained from live/dead staining, several cells were observed on the surface of the coated substrates for both CaP coating conditions. Once again, a greater number of cells was observed on the coatings prepared with Sr^2+^, supporting and consistent with the EIS results discussed above. Interestingly, the large flat plate-like dicalcium phosphate dihydrate particles observed on the coatings in Figure 5 are not present on the coatings after immersion in cell culture media for 7 days (Figure 11). The absence of the characteristic flat plate-like dicalcium phosphate dihydrate particles is most likely due to their increased solubility at physiological pH as compared to hydroxyapatite. Immersion in the culture medium during cell culture likely resulted in the dissolution of the dicalcium phosphate particles and re-precipitation as another CaP phase, most likely hydroxyapatite. The exact nature and composition of the coatings, with and without Sr^2+^, after immersion in solution for a period of time is to be determined in future dissolution studies. The increased corrosion protection combined with the likely osteoconductive characteristics of Sr^2+^ akin to Ca^2+^ is probably responsible for the increased cell attachment and proliferation observed on the Sr^2+^ containing CaP coatings.

Comparison of the live/dead results shown in Figure 9 and Figure 10 with a previous study that assessed the cytocompatibility of MC3T3-E1 cells cultured on biphasic hydroxyapatite and β-tricalcium phosphate coatings, with and without Sr^2+^, using the same coating method on Na_2_HPO_4_ pre-treated AZ31 substrates validates the positive effect Sr^2+^ has on cell proliferation and attachment [11]. Similar MC3T3-E1 cell densities were observed after 3 and 7 days of culture on the herein reported biphasic hydroxyapatite and dicalcium phosphate dihydrate coating, as well as the hydroxyapatite and β-tricalcium phosphate coatings. Furthermore, both studies revealed a large decrease in cell density as compared to the control group (tissue culture plastic) after 3 days of culture, which suggests that many of the MC3T3-E1 cells did not attach during initial cell seeding [11]. In terms of the current study, the decreased MC3T3-E1 cell attachment on the biphasic hydroxyapatite and dicalcium phosphate dihydrate coatings, with and without Sr^2+^, is likely related to the solubility of the coatings at the initial cell culture, time point. Hydroxyapatite is known to be insoluble at physiological pH, whereas dicalcium phosphate dihydrate exists as a highly soluble metastable phase and dissolves and reprecipitates as hydroxyapatite [1,39]. The rapid dissolution and reprecipitation behavior that dicalcium phosphate dihydrate demonstrates can create local areas of altered pH and ion concentrations which can lead to decreased cellular attachment, especially in dicalcium phosphate dihydrate rich portions of the coating. Once the dicalcium phosphate dihydrate converts to hydroxyapatite, the coating becomes much more stable, devoid of local pH and ion perturbations, and can facilitate the proliferation of the attached cells. Thus, incorporation of Sr^2+^ into the CaP coating increases the stability of the hydroxyapatite phase which results in a more stable coating consisting primarily of hydroxyapatite and facilitates increased initial MC3T3-E1 cellular attachment and proliferation. However, further in vitro dissolution studies which examine the phase composition transformation of the deposited coatings over time in a physiologically relevant solution must be conducted to confirm this, which was not the focus of the present study and will be a direction of future work.

In the current study, the coatings formed on AZ31 by MAO treatment prior to depositing CaP coatings were incapable of supporting MC3T3-E1 cell proliferation in comparison to the cells seeded either on tissue culture plastic or on CaP-coated AZ31. Interestingly, in other reports, similar MAO coatings have been shown to support either osteoblast or preosteoblast proliferation [18,28]. The incapability of MAO coatings in the current study to support cell proliferation may be due to the poor corrosion protection provided by these coatings in comparison to the CaP coatings (Figure 6 and Figure 7) or the increased corrosion of AZ31 in comparison to the alloys described in other studies. The porous nature of the MAO-treated coatings on AZ31 could also contribute to the poor corrosion protection, which also contributes to the lack of any MC3T3-E1 cell attachment and proliferation observed.

In comparing the two CaP coating conditions, the presence of Sr^2+^ in the CaP coatings was observed to support enhanced preosteoblast proliferation. Although Sr^2+^ has been demonstrated to support preosteoblast cell proliferation and differentiation, ionic concentrations in excess of 1 mM substantially reduced human mesenchymal stem cell proliferation [45]. The influence of Sr substitution on cell proliferation has also been studied by the authors in their previously published work (see reference [11]). In this work, the authors have clearly shown that increasing concentrations of Sr^2+^ beyond 5 and 10 percent followed by pretreatment to 350 °C showed lowered cell proliferation after 3 days of culture when treated with MC3T3-E1 and human mesenchymal stem cells (hMSCs) compared to coatings with no Sr^2+^ substitutions. This was deciphered by measuring the concentration of DNA. After 7 days of culture, a substantial increase in DNA concentration in comparison to uncoated AZ31 was observed for CaP-coated substrates without Sr^2+^ substitutions. However, there was minimal cell toxicity observed for both cell types on CaP-coated AZ31 substrates. Incorporation of Sr^2+^, however, also showed profound influence on osteogenic gene and protein expression. Therefore, it can also be confirmed that the presence of up to 11.5 mol. % Sr in CaP coatings is non-toxic to MC3T3-E1 cells and solutions prepared for CaP coatings with increased amounts of Sr^2+^ in comparison to those used in the current study (Table 2) may be of interest for future studies. These studies will be planned in the future. The results of these studies demonstrate that generation of these different cation-substituted CaP coatings on MAO-pretreated biodegradable Mg alloy substrates could offer unique opportunities for generating biodegradable Mg alloy scaffolds with improved and controlled corrosion for mineralized tissue engineering applications, as well as drugs and signaling molecule delivery systems.

## 4. Conclusions

MAO treatment was explored as a pretreatment technique on a biodegradable magnesium alloy prior to depositing CaP coatings either with or without Sr^2+^ substitution using an aqueous-based approach. Despite the presence of up to the nominal content of 10 mol. % Sr in the coating solution, a CaP coating consisting of a biphasic mixture of dicalcium phosphate dihydrate and poorly crystalline hydroxyapatite was formed. The presence of Sr^2+^ in the CaP coating was not observed to provide a substantial improvement in corrosion protection in comparison to the CaP coatings prepared without Sr^2+^ based on the results obtained using potentiodynamic polarization tests. However, the experimentally obtained and theoretically fitted data using equivalent circuit models from data collected by conducting EIS measurements highlights the positive effect Sr^2+^ has on increasing the corrosion resistance by providing an overall increase in charge transfer resistance. Interestingly, Sr^2+^-containing CaP coatings appeared to support enhanced MC3T3-E1 murine preosteoblast proliferation in comparison to both MAO-coated AZ31, and the CaP coatings prepared without Sr^2+^. Therefore, the feasibility of MAO as a pretreatment coating technique on AZ31 to facilitate the deposition of a corrosion resistant and cytocompatible Sr^2+^ containing CaP via solution immersion studied herein has been verified and warrants additional in-depth characterization. However, the current reported work is not without limitations and the additional studies that are planned in the future will aim to address these limitations by addressing the following: (1) Efforts will be made to decrease the overall porosity of the MAO coating by optimizing the MAO pretreatment parameters and conduct cross-sectional microstructural, morphological, and elemental analysis. (2) Thorough mechanical analysis will be performed to evaluate the adhesive strength of the MAO and CaP coatings. (3) The Sr^2+^ substitution within the calcium phosphate coatings and the apatite lattice will be rigorously quantified by using Rietveld Refinement. (4) In vitro dissolution studies will be conducted to assess the overall effect of Sr^2+^ substitution on the stability and phase transformation of the CaP coatings. (5) Sample sizes for all future testing will be increased to allow for a more statistically relevant and robust data analysis and assessment of coating variability. (6) In vitro quantitative cellular proliferation and protein expression analysis will be performed to further assess the effect of Sr^2+^ on cytocompatibility. (7) In vitro findings will be validated under in vivo conditions and the role of Sr^2+^ within the CaP coatings on biocompatibility and resorption capability will be assessed. Altogether, the formation of CaP coatings with increased amounts of Sr^2+^ on MAO-coated magnesium alloys and their capability to also support the differentiation of osteoblast progenitor cells towards a mature osteoblast lineage will be evaluated in future work and will be of much interest to the greater scientific community engaged in biomaterial research.

## Figures and Tables

**Figure 1 materials-18-04509-f001:**
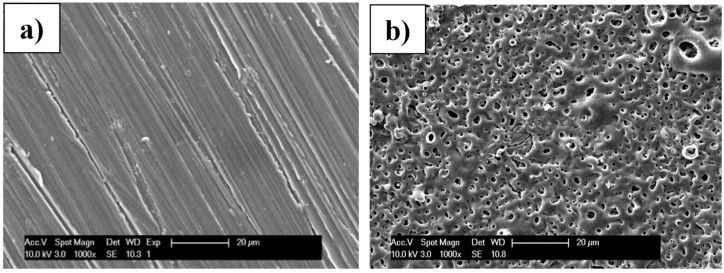
The microstructure of the (**a**) bare polished AZ31 substrate and (**b**) after micro-arc oxidation treatment. Images taken at 1000×.

**Figure 2 materials-18-04509-f002:**
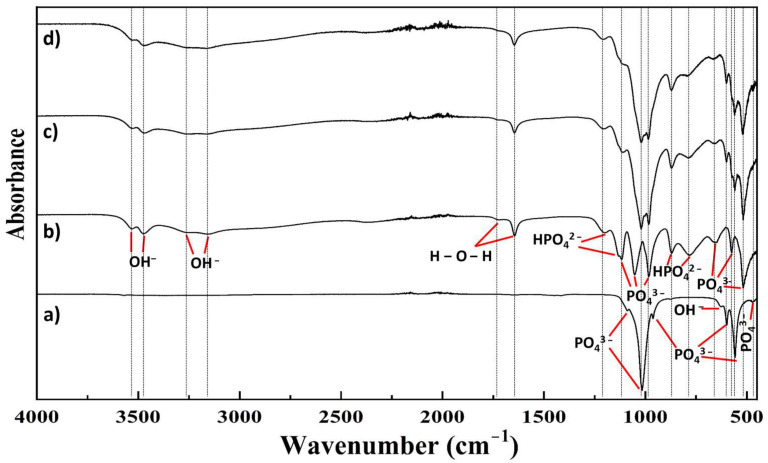
Fourier-transform infrared spectroscopy data in the range of 4000 cm^−1^ to 450 cm^−1^ collected from (**a**) commercially obtained hydroxyapatite (Sigma Aldrich, St. Louis, MO, USA), (**b**) commercially obtained dicalcium phosphate dihydrate (Sigma Aldrich, St. Louis, MO, USA), (**c**) calcium phosphate coatings on micro-arc oxidized AZ31 prepared without Sr (MAO-CaP), and (**d**) calcium phosphate coatings on micro-arc oxidized AZ31 prepared with Sr (MAO-SrCaP). Vertical dotted lines correspond to known characteristic bands of the hydroxyapatite and dicalcium phosphate dihydrate standard powders.

**Figure 3 materials-18-04509-f003:**
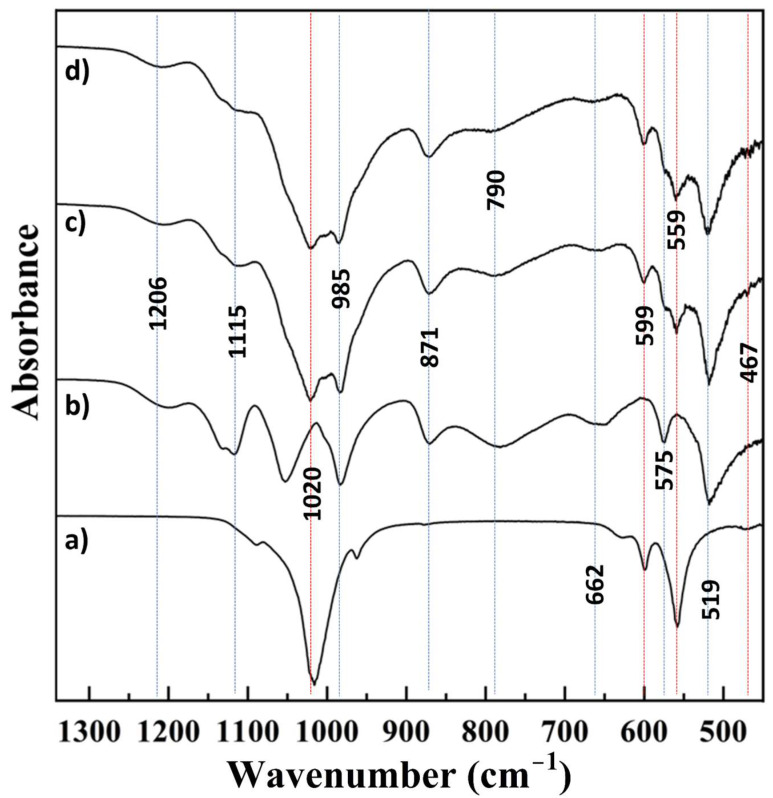
Fourier-transform infrared spectroscopy data in the range of 1350 cm^−1^ to 450 cm^−1^ collected from (**a**) commercially obtained hydroxyapatite (Sigma Aldrich, St. Louis, MO, USA), (**b**) commercially obtained dicalcium phosphate dihydrate (Sigma Aldrich, St. Louis, MO, USA), (**c**) calcium phosphate coatings on micro-arc oxidized AZ31 prepared without Sr (MAO-CaP), and (**d**) calcium phosphate coatings on micro-arc oxidized AZ31 prepared with Sr (MAO-SrCaP). The red and blue vertical lines represent hydroxyapatite and dicalcium phosphate dihydrate characteristic bands present in the coatings, respectively.

**Figure 4 materials-18-04509-f004:**
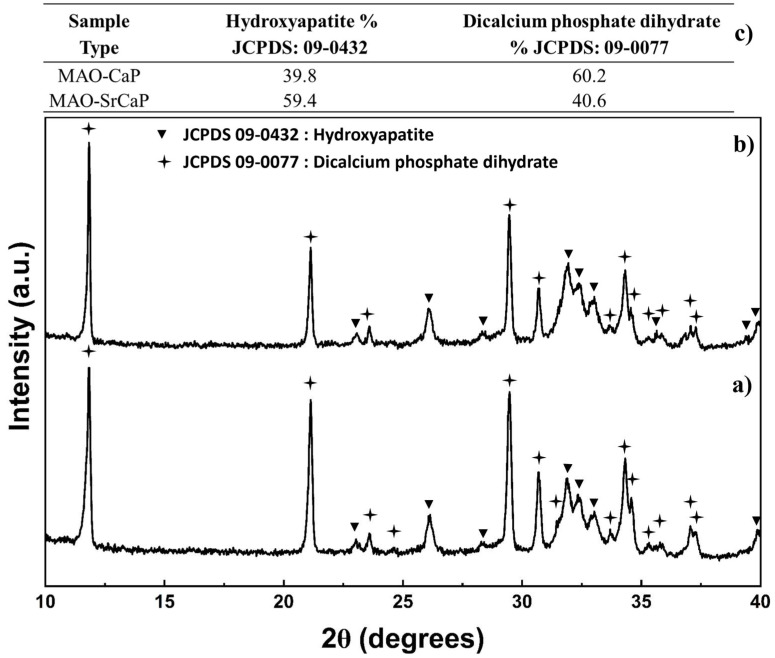
X-ray diffraction data collected from (**a**) calcium phosphate coatings on micro-arc oxidized AZ31 prepared without Sr (MAO-CaP), (**b**) calcium phosphate coatings on micro-arc oxidized AZ31 prepared with Sr (MAO-SrCaP), and (**c**) percent phase composition measured by Direct Derivation Method (DDM) semi-quantitative phase analysis.

**Figure 5 materials-18-04509-f005:**
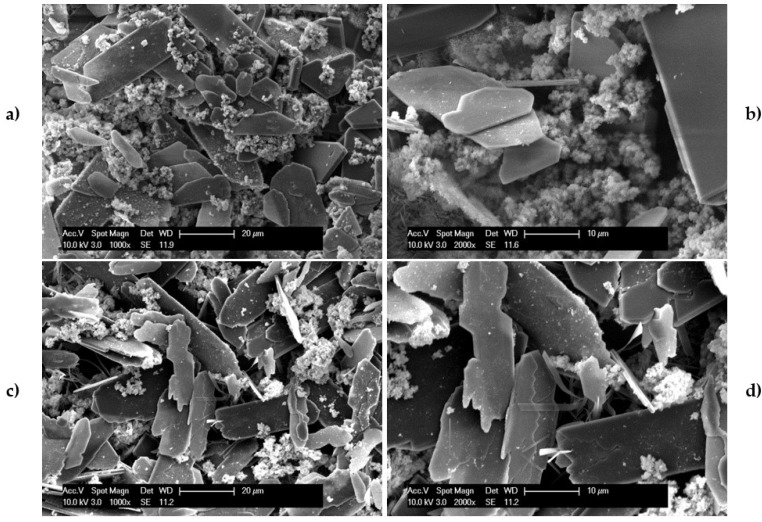
The microstructure of the AZ31 substrate surface after micro-arc oxidation treatment and the deposition of (**a**,**b**) calcium phosphate coatings without Sr (MAO-CaP) and (**c**,**d**) calcium phosphate coatings with Sr (MAO-SrCaP).

**Figure 6 materials-18-04509-f006:**
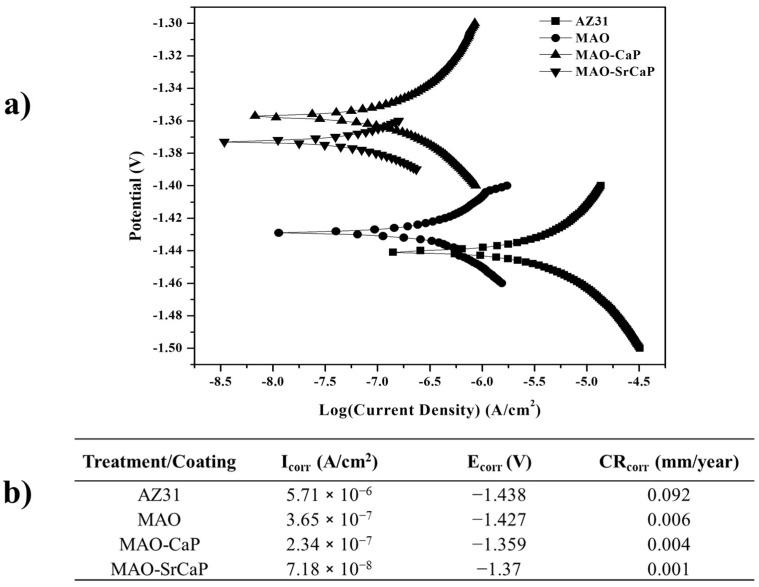
The potentiodynamic polarization data collected (**a**) from coated and uncoated substrates using Hank’s balanced salt solution maintained at 37 °C and (**b**) the Tafel extrapolated Ecorr and Icorr values. A scan rate of 0.001 V/s was used.

**Figure 7 materials-18-04509-f007:**
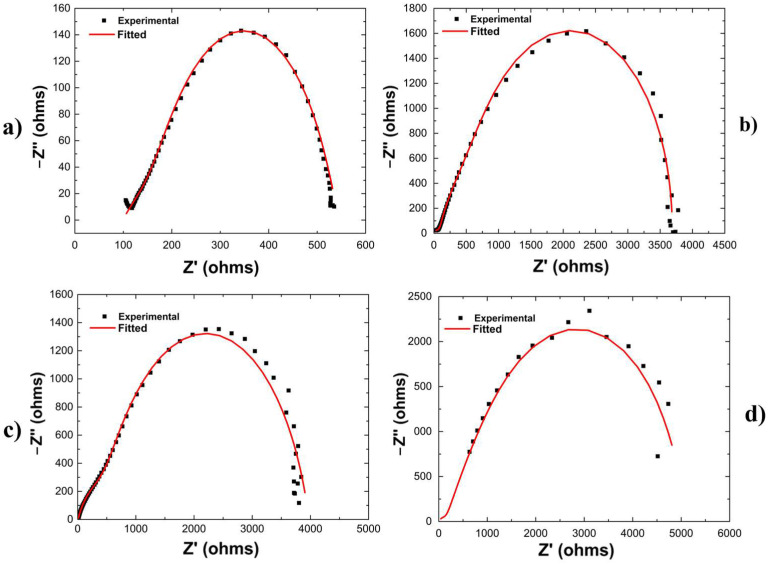
Experimentally obtained and theoretically fitted Nyquist plots obtained on (**a**) bare AZ31, (**b**) MAO-treated AZ31, (**c**) MAO-CaP-treated AZ31, and (**d**) MAO-SrCaP-treated AZ31.

**Figure 8 materials-18-04509-f008:**
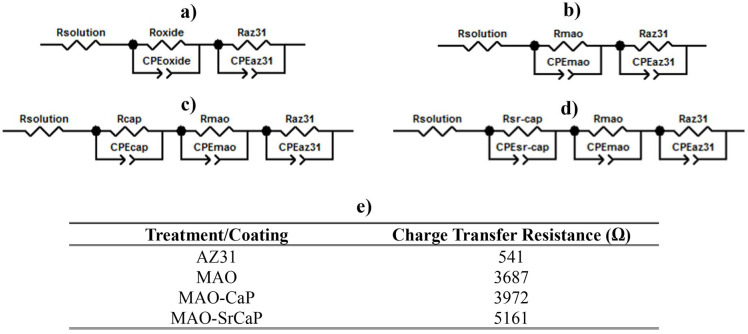
Representative equivalent circuit models for (**a**) bare AZ31, (**b**) MAO-treated AZ31, (**c**) MAO-CaP-treated AZ31, (**d**) MAO-SrCaP-coated AZ31 and (**e**) corresponding charge transfer resistance for each model system. CPE = constant phase element, R = resistance.

**Figure 9 materials-18-04509-f009:**
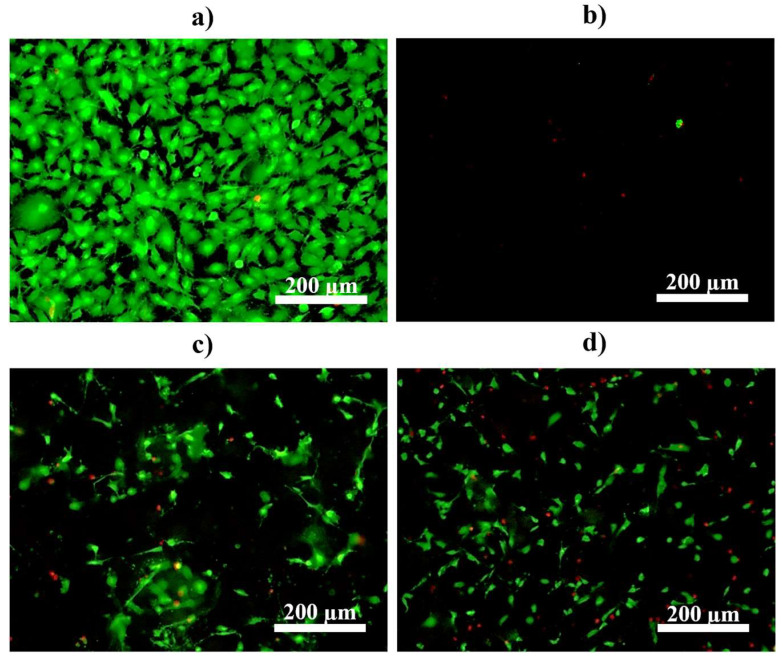
Live/dead staining using MC3T3-E1 cells after 3 days of culture in growth media on (**a**) tissue culture plastic, (**b**) micro-arc oxidized AZ31 (MAO), (**c**) calcium phosphate coatings without Sr (MAO-CaP), and (**d**) calcium phosphate coatings with Sr (MAO-SrCaP). Live cells are stained green and dead cells are stained red, respectively.

**Figure 10 materials-18-04509-f010:**
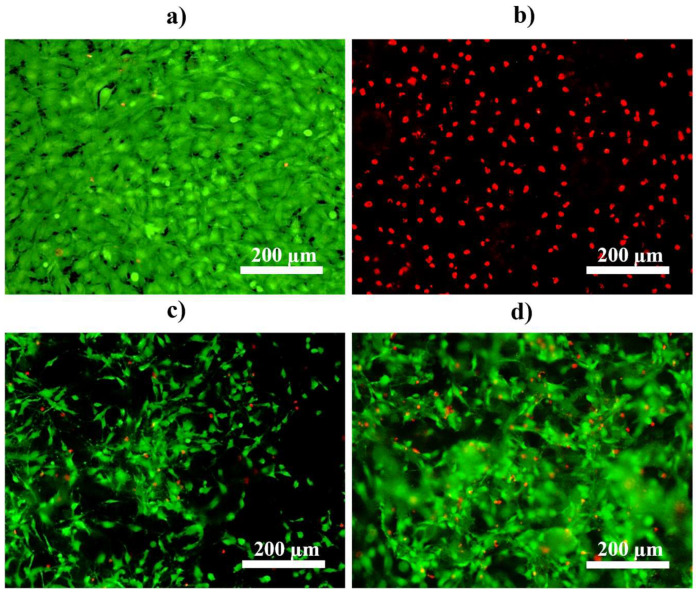
Live/dead staining using MC3T3-E1 cells after 7 days of culture in growth media on (**a**) tissue culture plastic, (**b**) micro-arc oxidized AZ31 (MAO), (**c**) calcium phosphate coatings without Sr (MAO-CaP), and (**d**) calcium phosphate coatings with Sr (MAO-SrCaP). Live cells are stained green and dead cells are stained red, respectively.

**Figure 11 materials-18-04509-f011:**
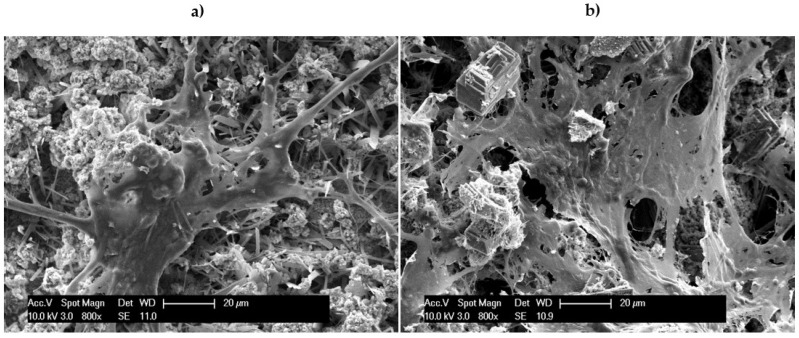
SEM images of MC3T3-E1 cells after 7 days on (**a**) calcium phosphate coatings without Sr (MAO-CaP), and (**b**) calcium phosphate coatings with Sr (MAO-SrCaP).

**Table 1 materials-18-04509-t001:** The composition of the two solutions that was used to deposit calcium phosphate coatings, with and without Sr, on micro-arc oxidized AZ31 substrates.

	CaCl_2_ (mM)	SrCl_2_ (mM)	Na_2_HPO_4_ (mM)
Sr0	100	-	60
Sr10	90	10	60

**Table 2 materials-18-04509-t002:** The elemental composition of the calcium phosphate coatings formed on micro-arc oxidized AZ31 substrates.

	mol. % Mg	mol. % Ca	mol. % Sr	mol. % P
Sr0	8.2	50.6	-	41.2
Sr10	9.7	43.2	11.5	35.6

**Table 3 materials-18-04509-t003:** Live and dead cell counts of MC3T3 cells after 3 days of culture in growth media on tissue culture plastic, micro-arc oxidized AZ31, calcium phosphate coatings without Sr, and calcium phosphate coatings with Sr. All measurements were performed in ImageJ.

Sample Type	Live Cells Counted	Dead Cells Counted	% Live Cells
Tissue culture plastic	402	3	99.2
MAO	3	8	11.1
MAO-CaP	104	18	85.2
MAO-SrCaP	182	12	93.8

**Table 4 materials-18-04509-t004:** Live and dead cell counts of MC3T3 cells after 7 days of culture in growth media on tissue culture plastic, micro-arc oxidized AZ31 (MAO), calcium phosphate coatings without Sr (MAO-CaP), and calcium phosphate coatings with Sr (MAO-SrCaP). All measurements were performed in ImageJ.

Sample Type	Live Cells Counted	Dead Cells Counted	% Live Cells
Tissue culture plastic	670	6	99.1
MAO	0	248	0
MAO-CaP	212	39	84.5
MAO-SrCaP	385	35	91.7

## Data Availability

The data sets presented in this article are not readily available at this time because the data also forms part of an ongoing study. However, the authors will be glad to share the relevant information upon request. Requests to access the data sets should be directed to pkumta@pitt.edu.

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
