# Peer review of "Feasibility Study of Strontium-Containing Calcium Phosphate Coatings on Micro-Arc Oxidized AZ31"

_materials, 2025, doi:10.3390/ma18194509_

Round 1

Reviewer 1 Report

Comments and Suggestions for Authors

Dear Editor and Authors,

This manuscript entitled "Feasibility Study of Strontium Containing Calcium Phosphate Coatings on Micro-Arc Oxidized AZ31" presents a relevant investigation into the development of strontium-doped calcium phosphate coatings on biodegradable AZ31 magnesium alloy substrates, pretreated via micro-arc oxidation (MAO). The work demonstrates a solid experimental design with promising implications for corrosion protection and cytocompatibility. However, I would recommend the authors address the following specific comments and suggestions to enhance the clarity, scientific rigor, and impact of the study.

  1. Please specify the novelty of this study. How does the proposed MAO pretreatment prior to Sr-CaP deposition provide advantages over previous strategies using hydrothermal or electrochemical methods? (e.g., compared to refs [11], [25]–[28].)
  2. Have the authors considered quantifying cell viability or metabolic activity (e.g., by MTT or Alamar Blue assay) to better understand the dose-dependent biological effects of Sr2+? This is very important for the text.
  3. Figures 8–10 provide qualitative cell imaging results; however, quantitative analysis (e.g., live cell counts, fluorescence intensity) is absent, limiting interpretation of biological performance. Please consider including bar charts or statistical summaries of cell proliferation on Day 3 and Day 7 across all conditions. This would greatly strengthen the biological conclusions. For the authors’ convenience, I am suggesting several recent publications that could be useful for strengthening the manuscript:
    1. Colloids and Surfaces B: Biointerfaces, 2024 – https://doi.org/10.1016/j.colsurfb.2024.114202
  4. FTIR and XRD results suggest Sr does not alter the CaP phase. However, no evidence is provided for whether Sr2+ enhances the chemical stability or dissolution resistance of the coatings in physiological media. Have the authors studied the dissolution behavior or ion release profile of the coating? A simple immersion test in HBSS with ICP analysis can provide insight into the long-term integrity of the coating.
  5. For potential clinical applications, the adhesion of the CaP coating is critical. The manuscript does not mention whether adhesion or mechanical integrity was evaluated.
  6. The manuscript is well-structured, but most cited references are relatively outdated, with few from the past three years. This limits the work’s connection to current research in calcium phosphate coatings and Mg alloy surface engineering. Thus, I recommend updating the Introduction and Discussion with more recent literature to improve relevance. Suggested references include:
    1. Coatings, 2023 – 10.3390/coatings13111874

Author Response

Response to Reviewer 1 can be found in the attached file. Please see the attached letter.

Reviewer 2 Report

Comments and Suggestions for Authors

Coating biodegradable Mg alloys with undoped and doped calcium phosphate is a promising route for degradation control and increasing the cytocompatibility of such alloys.

In continuation of previous works, Dr. P.N. Kumta, continuing to change the conditions for obtaining Strontium Containing Calcium Phosphate Coatings on AZ31, studied 2 samples of MAO treated AZ31, differing in the presence and absence of Sr in the composition with calcium phosphate.

despite the improvement in the corrosion resistance of alloys with the introduction of calcium phosphate coatings and strontium containing calcium phosphate coatings, cytocompatibility studies have not yet yielded high results.

It is known that an increase in the strontium content in coatings leads to a decrease in the degradation rate of alloys. The resulting lower concentration of Mg in the surrounding extracts will contribute to a decrease in the cytotoxicity of the material. In this regard, a general question arises regarding the contents of the manuscript: why did the authors studied a sample with only one Sr content?

Some specific comments are the following:

In the Highlights, in the phrase

“Sr doped CaP coatings show improved MC3T3-E1 cell attachment and proliferation”

the result is described too briefly, which may lead to its misunderstanding, since improved MC3T3-E1 cell attachment and proliferation is observed in comparison with a similar material without Sr or for a sample without coating at all and is still far from successful MC3T3-E1 cell attachment and proliferation for other available materials.

Abstract:

“Calcium phosphate coatings with up to 11 mol. % Sr were formed on micro-arc oxidized AZ31 substrates.”

The indication in the introduction of 11% mol. % Sr may cause misunderstanding by the reader, since it exceeds its initial load. Even though the authors explain this by partial dissolution of Ca in their results, it is recommended that the authors provide a reliable amount of added strontium in their abstract.

Experimental Section:

Page 4: The authors write “The coatings were, accordingly, deposited for 5-, 10-, or 20-minute periods,” but in the following section 2.3. Calcium Phosphate Coating Formation they do not specify which specific treated alloy sample was used.

Page 5: Please provide an explanation for the abbreviation PBS when first mentioned.

Results:

Page 5: The authors write “MAO treatments were performed on polished AZ31 substrates for a period of ten minutes,” but provide justification for choosing this time only for the longer period (20 min), but do not explain not using the 5 min period.

Please check whether the following sentence actually refers to pores of 1 mm in size or does it mean the depth of the pores: The pores detected in the current work were generally much less than 5 µm in diameter being the largest size and as small as ~1 mm. With such large holes, the coating can hardly be called homogeneous.

Page 6: Please provide in the experimental section the sources of commercially available pure hydroxyapatite and pure dicalcium phosphate dihydrate used for FTIR comparison.

Page 7. The authors discuss how pH could affect the phase composition of the coating formed on the alloy surface, but do not provide the results of a simple experiment to determine the pH of the reaction solution, which is, of course, a drawback of this discussion.

In the introduction, the authors write about homogeneous oxide coatings formed after MAO treatment compared to samples obtained by other methods, however, in Fig. 2, the authors present essentially the same images, differing only in the degree of magnification by 2 times. I would recommend providing only one image of AZ31 substrate after MAO treatment, but adding either an image of the alloy surface before MOA, or an example of heterogeneity of alloy treated with other common methods, to demonstrate the real advantage of the MAO method. As I see from the authors' previous publications, acid treatment leads to the formation of a very rough surface layer, but nevertheless quite homogeneous. Therefore, from the point of view of surface modification, the advantages of the MAO method, based on the data provided in the manuscript, are not yet obvious to me. Perhaps, the interpretation of the treatment results should be built not from the point of homogeneity, but from the point of view of roughness.

Page 9: What is the measurement error of the elemental analysis of strontium? Could the overestimated value be the result of an error? And what was the standard deviation of the three measurements (the authors write about each experiment being performed three times)?

Please change the caption of Figure 4 for greater clarity. It is difficult to make sense of several “and” (…and c) and d)), without referring to the text of the manuscript.

The authors do not write anything about the strength of the coating adhesion to the alloy. How strong is it at least qualitatively? For example, in Fig. 10 I do not see any traces of platelet morphology, characteristic of strontium containing calcium phosphate coatings on AZ31. At the same time, based on Fig. 9c, it is hardly worth assuming that the density of living cells is high enough to completely cover the coating.

The expression "Once again" occurs frequently on page 16

References:

Numbering is duplicated.

Author Response

Comments for reviewer 2 have been addressed. The comments can be found in the attached file.

Reviewer 3 Report

Comments and Suggestions for Authors

While the research topic is relevant and the experimental methodology covers a wide range of characterization techniques, the manuscript presents critical flaws in both the structural interpretation and analytical rigor, which undermine the validity of its conclusions. The paper  requires full revision, as it is based on unfounded and unproven claims. 

  1. Although the entire paper aims to demonstrate the formation and influence of calcium phosphate (CaP) phases, including those doped with Sr²⁺, the XRD analysis presented in the article is brief and lacking in depth. The phases detected by XRD are not clearly identified. It is stated that dicalcium phosphate dihydrate (brushite) and hydroxyapatite (HA) were formed, but the positions and attributions of the diffraction peaks (2θ) are not explicitly indicated, nor is it directly correlated with the standard JCPDS files.

2. In the XRD pattern, no labels appear for the characteristic peaks and the corresponding phases are not marked. This omission makes it impossible to verify the statements regarding the phases present.

3. No quantitative analysis is provided (Rietveld refinement, percentage estimation or evaluation of the degree of crystallinity).

4. Furthermore, no visible effect of Sr doping on the diffraction patterns is observed, which partially contradicts subsequent statements about possible ionic substitutions.This lack of rigor seriously affects the interpretation of the entire set of results, because:

5. All conclusions regarding phase composition, possible Sr²⁺/Ca²⁺ substitution, and impact on electrochemical and biological properties are based on this supposed XRD phase identification.

Without a convincing XRD characterization, the interpretation of FTIR, SEM, and electrochemical results becomes indirect and speculative.

6. The band at ~1650 cm⁻¹ is incorrectly attributed to "brushite" by the authors. In reality, it corresponds to the δ(H–O–H) vibration mode of physisorbed water molecules and is not sufficient evidence for the dicalcium phosphate dihydrate phase.

7. The characteristic bands of the phosphate ion (PO₄³⁻), which are essential for the identification of HA or DCPD, are not rigorously described.

8. It is not mentioned whether the ν₁ ~960 cm⁻¹ band, specific to hydroxyapatite, is present or not. This directly affects the conclusion regarding the "presence of weakly crystalline HA".

Notably, the phase identification based on XRD and FTIR is insufficient, incomplete, and at times incorrect, yet it forms the foundation upon which all subsequent interpretations (electrochemical, morphological, biological) are based. Without proper phase confirmation, the manuscript's claims about Sr incorporation, biphasic coatings, and phase-dependent corrosion or cytocompatibility effects become speculative.

Therefore, despite the interesting concept and adequate experimental breadth, the manuscript in its current form does not meet the standards of scientific rigor required for publication in Materials.

Author Response

Comments to Reviewer 3 can be found in the attached file. Please refer to the attached file.

Round 2

Reviewer 1 Report

Comments and Suggestions for Authors

The authors have addressed all what I concerned, thus the manuscript can be accepted now.

Reviewer 2 Report

Comments and Suggestions for Authors

Please, correct the following:

Line 258 – FTIR in Fig. 3 is from 1350 cm-1
Line 486 Na2HPO4 – subscript for 4

Reviewer 3 Report

Comments and Suggestions for Authors

Dear authors,

  1. It is stated that there are many works on CaP-Sr, but the manuscript does not state clearly enough why the MAO + Sr-CaP approach is significantly different from previous ones.
  2. The methodology is detailed, but there are some weaknesses:

    2.1 It is not clear why only the 10-minute MAO treatment was chosen, when other durations were also mentioned.

    2.2 Explanations regarding the number of replicates and statistical analysis are missing (although the final section states that testing was done in triplicate).

    2.3 In cytocompatibility experiments, the comparison is made only with the culture plastic and the AZ31 substrate, without including an additional control (e.g. CaP deposited by another method).

  3. The inclusion of Sr in the CaP structure is mentioned, but not rigorously quantified (it is only stated as "~11.5 mol%", without a detailed analysis of the ionic substitution).
  4. Electrochemical tests (EIS and PDP) show improvements, but comparison with other studies in the literature is only spot-on, not systematic and under the same conditions.

    5. In vitro experiments with MC3T3-E1 confirm better proliferation on Sr-CaP, but the proliferation level remains well below that of culture plastic – this fact is not critically discussed.

    6. The discussion confirms the improvement of properties by Sr, but does not explain in detail the mechanisms by which Sr increases corrosion resistance (e.g. structural stability of doped HAp.. 

     7. The limitations of the method are not sufficiently discussed: porosity of the MAO layer, possible variability of the composition, lack of validation under in vivo conditions. Please complete.

    8. There is no critical comparison between the proposed method and other existing pretreatment + CaP methods (e.g. hydrothermal, electrochemical). Please complete the discussion.
